# The structural and functional investigation into an unusual nitrile synthase

Hao Li[1,4], Jian-Wen Huang[1,4], Longhai Dai [1,4], Haibin Zheng[1], Si Dai[1], Qishan Zhang[1], Licheng Yao[2], Yunyun Yang[1], Yu Yang [1], Jian Min [1], Rey-Ting Guo [1,3] ✉ & Chun-Chi Chen [1,3] ✉

The biosynthesis of neurotoxin aetokthonotoxin (AETX) that features a unique structure of pentabrominated biindole nitrile involves a first-of-its-kind nitrile synthase termed AetD, an enzyme that shares very low sequence identity to known structures and catalyzes an unprecedented mechanism. In this study, we resolve the crystal structure of AetD in complex with the substrate 5,7-di-Br-L-Trp. AetD adopts the heme oxygenase like fold and forms a hydrophobic cavity within a helical bundle to accommodate the indole moiety. A diiron cluster comprising two irons that serves as a catalytic center binds to the carboxyl O and the amino N of the substrate. Notably, we demonstrate that the AetD-catalyzed reaction is independent of the bromination of the substrate and also solved crystal structures of AetD in complex with 5-Br-L-Trp and L-Trp. Altogether, the present study reveals the substrate-binding pattern and validates the diiron cluster-comprising active center of AetD, which should provide important basis to support the mechanistic investigations into this class of nitrile synthase.

Aetokthonotoxin (AETX) is an unusual pentabrominated biindole nitrile produced by *Aetokthonos hydrillicola*, an epiphytic cyanobacterium that colonizes on the invasive aquatic plant *Hydrilla verticillata*[1]. AETX is a neurotoxin that was recently identified as a causative agent of an epizootic event in the United States known as avian vacuolar myelinopathy (VM). Wild birds that prey on waterbirds and other animals that consume the bacteria-colonized plants may succumb to VM. The association of AETX and VM has been demonstrated on nematodes, zebrafish, and leghorn chicken[1]. Although AETX-mediated effects on mammals remain unclear, the toxin that traverses the food chains is a potential health threat to the human population.

AETX represents the first naturally produced 1,2′-bi-1*H*-indol, which possesses unique chemical features including the five-bromo substituents, the N1-C2′ linkage between indole moieties and

the substructure of 5,7-di-Br-indole-3-carbonitrile (Fig. 1a). Breinlinger et al. identified AETX biosynthetic gene cluster that comprises six coding genes (AetA to AetF)[1], and five of them (except AetC) have been proven required for the AETX synthesis in vitro[2]. The AETX biosynthesis process is initiated from L-Trp (3), which is brominated on C5 position by a flavin-dependent tryptophan halogenase termed AetF to yield 5-Br-L-Trp (2). Then the reaction bifurcates into two routes: (i) 2 is converted to 5-bromoindole by a PLP-dependent tryptophanase termed AetE and then brominated by AetA on two sites of the pyrrole ring to yield 2,3,5-tribromoindole; (ii) 2 is brominated again by AetF on C7 to yield 5,7-di-Br-L-Trp (1), which is subsequently converted by a nitrile synthase termed AetD to yield 5,7-di-Br-indole-3-carbonitrile (4). Finally, a cytochrome P450 termed AetB catalyzes the biaryl coupling of 2,3,5-tribromoindole and 4 to afford AETX.

[1]State Key Laboratory of Biocatalysis and Enzyme Engineering, Hubei Hongshan Laboratory, Hubei Collaborative Innovation Center for Green Transformation of Bio-Resources, Hubei Key Laboratory of Industrial Biotechnology, School of Life Sciences, Hubei University, Wuhan 430062, PR China. [2]Hubei Gongtong Steroid Drug Research Institute, Wuhan 430073, PR China. [3]Department of Immunology and Pathogen Biology, School of Basic Medical Sciences, Hangzhou Normal University, Hangzhou 311121, PR China. [4]These authors contributed equally: Hao Li, Jian-Wen Huang, Longhai Dai. ✉e-mail: guoreyting@hubu.edu.cn; ccckate0722@hubu.edu.cn

## a

### "AetD-mediated pathway"

## b

### "Aldoxime-nitrile pathway"

## c

### "ToyM-mediated pathway"

**Fig. 1 | Biosynthetic pathways of natural nitrile compounds. a** AetD-mediated pathway in the AETX biosynthetic pathway. **b** Aldoxime-nitrile pathway. **c** ToyM-mediated pathway. The sources of N atoms of the nitriles are colored in red.

The AetD-catalyzed nitrile functionalization is particularly fascinating. The current knowledge of the biosynthesis of nitrile-containing natural compounds is limited. One characterized route that involves aldoxime formation and dehydration requires at least three separate enzymes[3–5] (Fig. 1b). The other one catalyzed by an ATP-dependent nitrile synthetase termed ToyM, which transforms carboxylic acid to nitrile by using exogenous ammonia as the source of nitrogen[6,7] (Fig. 1c). The AetD-mediated nitrile construction operates via a mechanism that is different from these currently acknowledged ones. Akak et al. conducted biochemical analyses and confirmed that Fe(II) is required for AetD activity and the nitrogen atom in the nitrile is sourced from the α-amino group of the substrate instead of exogenous nitrogen or the enzyme residue (Fig. 1a)[2]. Altogether, AetD is an unprecedented nitrile synthase that acts in more atom economic fashion in comparison with other known nitrile synthetic enzymes. Unfortunately, it is difficult to gain further insights into the molecular mechanism of AetD as this enzyme shares a very low sequence identity with characterized proteins and lacks any conserved domains.

In this work, the crystal structures of AetD in complex with its substrate and a diiron cluster are reported. In addition, we also found that the AetD-catalyzed reaction is independent of the halogenation of the substrate. These results should be of great importance to guide further mechanistic investigations of AetD.

## Results and discussion

### Crystal structure of AetD

We failed to grow the apo-form crystal of AetD but successfully grew co-crystals of AetD and **1** and eventually solved the complex structure (Table 1). This complex was termed AetD/1/Fe because it was obtained by soaking with $(NH_4)_2Fe(SO_4)_2$. AetD/1/Fe contains two polypeptide chains in an asymmetric unit, which appear to align as a homodimer with a buried interface estimated to 1656.5 $Å^2$ that comprises helix α4b, partial α2, and the intertwined helix α4a (Fig. 2a). In order to validate

the structural observations, a size exclusion chromatography analysis was conducted. The results are consistent with the structural finding that the molecular mass of AetD in solution corresponds to a dimer (Supplementary Fig. 1, monomer and dimer molecular weight, 27.97 and 55.94 kDa, respectively).

The monomeric AetD polypeptide folds into seven antiparallel α-helices (α1-α7) that are connected through short loops (Fig. 2b). The continuous polypeptide chain of each molecule of AetD can be observed except a missing fragment between helix α6 and α7 (residue 180-187 are missing in AetD/**1**/Fe) (Figs. 2a and 2b), which is also not seen in other AetD complex structures reported in the present study (see below). This is presumed a result of local flexible conformation and an alternative crystal packing environment might be required to produce otherwise results.

AetD shares a very low sequence identity to currently characterized proteins, thus we performed the searching with the DALI server to identify known structures that adopt consensus folding. The top ranking homologous structures of the searching results belong to 'heme-oxygenase-like' fold (HO-like fold), including a nonheme diiron oxidase UndA[8,9], a nonheme diiron oxidase BesC[10], an uncharacterized TenA homolog from *Pseudomonas syringae* (PDB ID, 3OQL) and the *Chlamydia* protein associated with death domain (CADD)[11] (Supplementary Fig. 2a). UndA and BesC are oxidases that harbor a diiron cluster as a catalytic center to exercise decarboxylation and C-C bond cleavage, respectively[9,10], whereas TenA is a protein with unknown function. CADD has been recently confirmed as a *p*-aminobenzoate synthase that hires a heterobimetallic Fe:Mn cofactor to exhibit its optimal activity[12,13], although a diiron cluster was observed in the enzyme in crystallographic analyses[11]. Another protein on the list that carries the diiron scaffold is an *N*-oxygenase termed SznF, whose HO-like domain is flanked by an *N*-terminal domain and a cupin domain (Supplementary Fig. 2b). Despite sharing very low sequence identity, AetD, and these HO-like diiron proteins are organized following the

**Table 1 | Data collection and refinement statistics for AetD complex structures**

| Bound ligand<br>PDB code | AetD/1/Fe<br>5,7-Di-Br-L-Trp, Fe<br>8JI3 | AetD/2/Fe<br>5-Br- L-Trp, Fe<br>8JI5 | AetD/3/Fe<br>L-Trp, Fe<br>8JI7 | AetD/1<br>5,7-Di-Br-L-Trp<br>8JI2 | AetD/2<br>5-Br- L-Trp<br>8JI4 | AetD/3<br>L-Trp<br>8JI6 |
|---|---|---|---|---|---|---|
| **Data collection** | | | | | | |
| Space group | $P4_1$ | $P4_1$ | $P4_1$ | $P4_1$ | $P4_1$ | $P4_1$ |
| Unit cell | | | | | | |
| a, b, c (Å) | 65.5, 65.5, 117.1 | 65.6, 65.6, 117.3 | 65.4, 65.4, 117.3 | 65.2, 65.2, 116.9 | 65.4, 65.4, 116.6 | 65.3, 65.3, 116.1 |
| α, β, γ (°) | 90, 90, 90 | 90, 90, 90 | 90, 90, 90 | 90, 90, 90 | 90, 90, 90 | 90, 90, 90 |
| Resolution (Å)[a] | 25.00-1.78<br>(1.84-1.78) | 25.00-2.01<br>(2.08-2.01) | 25.00-1.61<br>(1.67-1.61) | 25.00-1.75<br>(1.81-1.75) | 25.00-1.67<br>(1.73-1.67) | 25.00-1.90<br>(1.97-1.90) |
| No. of observed reflections | 47258 (4708) | 32155 (3018) | 63234 (6318) | 48871 (4905) | 56389 (5623) | 37037 (3756) |
| Redundancy | 4.5 (4.8) | 3.9 (3.3) | 5.0 (5.3) | 4.3 (4.6) | 4.3 (4.6) | 3.4 (3.4) |
| Completeness (%) | 99.7 (100.0) | 97.6 (92.5) | 99.5 (99.8) | 99.7 (100.0) | 99.7 (100.0) | 96.1 (97.4) |
| Average $I/\sigma$ (I) | 25.1 (2.2) | 36.8 (3.2) | 38.3 (2.7) | 28.4 (2.1) | 22.1 (2.2) | 33.0 (2.4) |
| $R_{merge}$ (%)[b] | 5.3 (67.5) | 2.8 (48.8) | 3.7 (62.3) | 4.3 (64.8) | 5.6 (46.5) | 2.8 (49.9) |
| CC 1/2 | 0.991 (0.835) | 0.998 (0.822) | 0.996 (0.845) | 0.999 (0.808) | 0.996 (0.862) | 0.999 (0.827) |
| **Refinement[c]** | | | | | | |
| $R_{work}$ (%) | 19.3 | 20.6 | 22.1 | 17.1 | 17.2 | 17.4 |
| $R_{free}$ (%) | 24.4 | 27.3 | 27.6 | 21.9 | 20.7 | 22.7 |
| r.m.s.d. bonds (Å) | 0.010 | 0.008 | 0.011 | 0.011 | 0.011 | 0.008 |
| r.m.s.d. angles (°) | 1.591 | 1.515 | 1.663 | 1.673 | 1.719 | 1.477 |
| **Ramachandran statistics** | | | | | | |
| Most favored (%) | 98.9 | 98.9 | 98.9 | 98.9 | 99.6 | 99.1 |
| Allowed (%) | 1.1 | 1.1 | 1.1 | 1.1 | 0.4 | 0.9 |
| Outliers (%) | 0 | 0 | 0 | 0 | 0 | 0 |
| **Average B-factor (Å²)** | | | | | | |
| Protein/atoms | 43.9/3793 | 32.7/3808 | 21.1/3803 | 39.6/3886 | 34.2/3909 | 25.1/3878 |
| Water/atoms | 44.7/200 | 32.3/194 | 31.3/564 | 45.6/292 | 41.4/324 | 32.4/387 |
| Ligand/atoms | 33.0/39 | 23.9/37 | 13.8/35 | 43.2/66 | 29.9/55 | 19.9/45 |

[a]Values in parentheses are for the highest resolution shell.

[b]$R_{merge} = \Sigma_{hkl}\Sigma i \, | \, I_i(hkl)- \langle I(hkl)\rangle \, |/\Sigma_{hkl}\Sigma_i I_i(hkl)$, in which the sum is over all the $i$ measured reflections with equivalent miller indices $hkl$; I($hkl$) is the averaged intensity of these $i$ reflections, and the grand sum is over all measured reflections in the data set.

[c]All positive reflections were used in the refinement.

same topology, suggesting that AetD should adopt an overall structure reminiscent of HO-like fold (Supplementary Fig. 2b).

## AetD and substrate interaction network

In the complex structure of AetD/**1**/Fe, the electron density maps that can be unambiguously modeled with a 5,7-di-Br-L-Trp was clearly observed in a cavity formed within the helical bundles (Fig. 2b, c and Supplementary Fig. 3). The bromoindole part of **1** is surrounded by several hydrophobic residues with the indole ring clamped by I146 and F44. In addition, the side chain of two polar residues, Y167 and S214, are within hydrogen bond distance to the carboxyl O atom and Br on C7, respectively (Fig. 2c). In addition to the substrate, additional maps that should correspond to two metal ions were observed adjacent to **1** (Fig. 2c). Based on the facts that Fe(II) is required for AetD activity[2] and that other HO-like protein structures (except BesC) carry a diiron cluster in the corresponding region (Supplementary Fig. 2b), we modeled two Fe ions in the complex of AetD/**1**/Fe. The Fe1 is coordinated by protein residue H79, H172, E176, a nearby water molecule, and the N and O atoms of **1**, and the Fe2 by D76, E140, E176 and H179 (Fig. 2d).

The type of metal ions modeled in the AetD/**1**/Fe structure could be complicated by the presence of Ni ions in the purification procedures (nickel column) and crystallization buffer. This event can be resolved by using anomalous diffraction technology to directly identify the species of the metal ion observed in crystals. However, it is hard to determine whether the signal of Ni is derived from the ions in the crystals or in the crystallization buffer. We thus performed atomic absorption spectrometry and detected signal of Fe ion and not for Ni ion in AetD protein (Supplementary Fig. 4), indicating that the purified AetD protein only carries Fe ion. Then we tried to grow crystals in buffers free of Ni ion but the efforts eventually failed. Nonetheless, we successfully solved the structure of AetD/1 crystal that was not soaked with Fe ion. The AetD/1 structure harbors the substrate and a metal ion that is coordinated and located as the Fe1 observed in AetD/1/Fe (Supplementary Fig. 5). Based on the fact that the purified AetD carries Fe ion and the highly identical coordination status to other HO-like proteins, this metal ion should be reliably modeled with a Fe ion. However, it should be noted that the possibility that a heterogeneous mixture containing Fe and Ni ion was observed in these crystals cannot be completely excluded. In addition, the presence of both $Fe^{2+}$ and $Fe^{3+}$ is possible although the crystallization and soaking trials were conducted under aerobic conditions. Intriguingly, the Fe2-corresponding site is vacant and the Fe2-coordination networks are dissociated owing to the disordered helix α6 that deviates H179 away from the Fe2 site (Supplementary Fig. 5). Because AetD/**1** and AetD/**1**/Fe share the same unit cell and space group, the conformational change of helix α6 is not a result of alternative crystal packing. Instead, it is possible that Fe ion in the soaking trial could bind to the enzyme and stabilize H179 to constitute the intact diiron cluster. Notably, a metal ion that is located distant from the Fe2 site and coordinated by waters, a TRIS and/or residues including E140 and H72 was observed in AetD/**1** (Supplementary Fig. 5). This metal was modeled with a Ni ion, whose presence could be competed by soaking crystals with Fe ion.

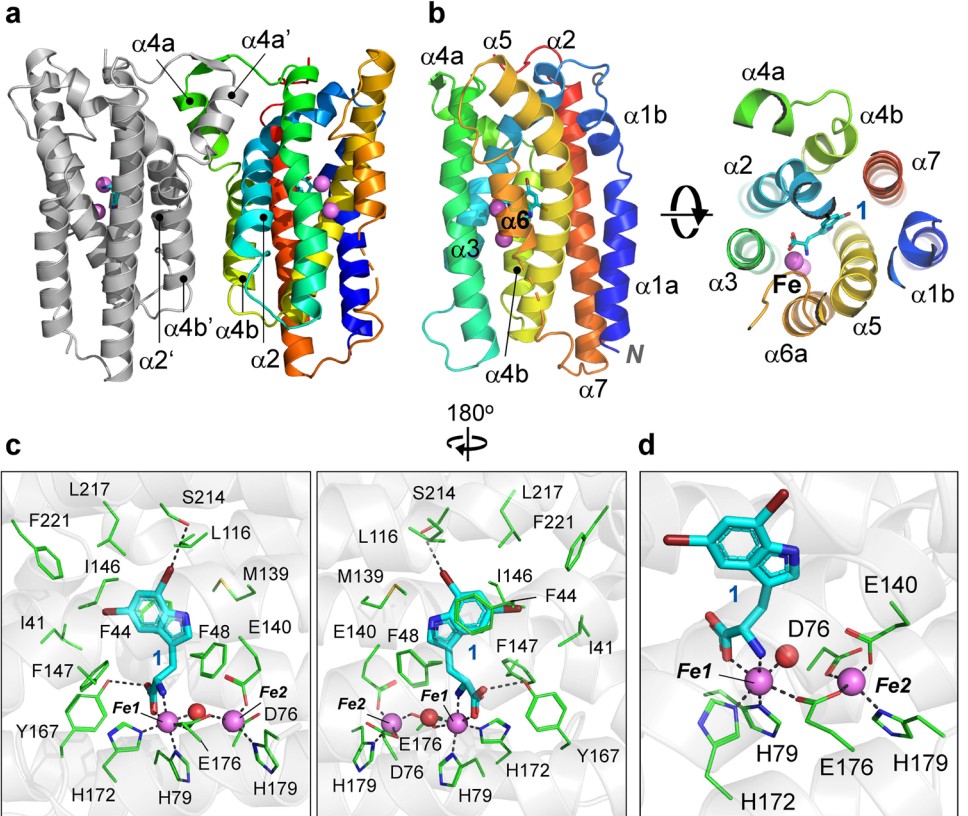

**Fig. 2 | The crystal structure of AetD in complex with the substrate. a** The two polypeptide chains in an asymmetric unit of AetD/**1**/Fe complex are displayed in cartoon models with one chain colored in gray and the other in rainbow. **b** The overall structure of monomeric AetD. The protein structure is presented by a cartoon model with each helices denoted numerically. Two views are related by 90° at X-axis. Compound **1** and two Fe ions are shown in stick and sphere, respectively. **c** The enzyme-substrate interaction network. The bound substrate and Fe ions are displayed as described in (**b**), with two Fe irons labeled numerically. Fe-coordinating water molecule is shown in a red sphere, and the protein residues that constitute the substrate-binding pocket are shown in line models. The $2F_o$-$F_c$ and $F_o$-$F_c$ maps of **1**, Fe ions and the coordinating water and residues are shown in Supplementary Fig. 3. Two views are related by 180° at Y-axis. **d** The Fe ions coordination status. The Fe ions, water molecules, and protein residues are shown as described in (**c**). The dashed lines indicate distance <3.5 Å.

## AetD-catalyzed reaction is independent of bromination of 1

In AetD/**1**/Fe, the carboxyl and amino group of **1**, where the nitrile functionalization takes place, participate in the Fe1 coordination, whereas the indole moiety binds in the nearby cavity. It is thus reasonable to suspect that the diiron-containing machinery involves in the nitrile construction while the bromination on the indole moiety might not be a prerequisite feature for the AetD substrate. To address this issue, we assembled AetD reactions that contained 5-Br-L-Trp (**2**) and L-Trp (**3**) as substrates. We detected products with molecular mass consistent to 5-Br-indole-3-carbonitrile ([M + H] $m/z$, 220.9684; theoretical molecular mass, 219.96, Supplementary Fig. 6) and 3-cyanonitrile ([M + H] $m/z$, 143.0602; theoretical molecular mass, 142.05, Supplementary Fig. 6), respectively (Fig. 3a). These results clearly demonstrate that AetD can transform L-Trp that carries single or no bromination, though lower transformation efficacy was observed for **3** (substrate conversion rate, ~35% for **3** and 100% for **1** and **2**). We also solved the crystal structure of AetD in complex with **2** and **3**, which reveal that these compounds bind in the pocket as that was observed in AetD/**1**/Fe. Their indole moieties were found to bind in a pattern essentially identical to that of **1**, with the carboxyl and amino groups also coordinating the Fe1 ion (Fig. 3b and Supplementary Fig. 7). Notably, 2 lacks the S214-mediated interaction and 3 forms even fewer contacts to the enzyme in comparison with 1 and 2 (Fig. 3b). This could be one factor accounting for the lower transformation efficacy of 3. Altogether, these results indicate that the AetD-catalyzed nitrile synthesis is independent of the presence of halogenation on L-Trp. In addition, the structures of these complex crystals without soaking with

Fe ion were also solved (Table 1). These complexes harbor the same metal ion coordination statuses as that was observed in the AetD/1 structure (Supplementary Fig. 8), which provide additional evidences to support the conclusions regarding the formation of the diiron cluster in AetD/Fe complex crystals.

## Diiron cluster of AetD and implications in catalytic mechanism

Our complex structures indicate that AetD harbor a diiron cluster analogous to those in other HO-like structures including UndA, CADD, and SznF (Fig. 4a). No electron density map that corresponds to metal ion was seen in the currently reported BesC structure but the existence and requirement of diiron-peroxo species of this enzyme has been validated[10,14]. In stark contrast to the low sequence identity (<11%), the Fe-coordinating ligands in these HO-like proteins are highly conserved (Fig. 4a). Intriguingly, although Fe1 is coordinated by two His and one Glu, their order of appearance in the primary sequence of AetD (H-H-E) is different from those in other HO-like proteins (E-H-H) (Supplementary Fig. 9). This complicates the identification of Fe1-coordinating residues only based on the sequence alignment. To further validate the structural observations, we thus conducted mutagenesis experiments to probe the role of the Fe-coordinating residues of AetD. As shown in Fig. 4b, all variants that harbor Ala substitution of Fe-coordinating residues lost catalytic activity, indicating the significance of diiron cluster in the action of AetD.

Based on the presented structural analyses and mutagenesis experiments, AetD should employ a diiron cluster as a catalytic center, a feature shared by HO-like diiron enzymes. Therefore, the AetD-

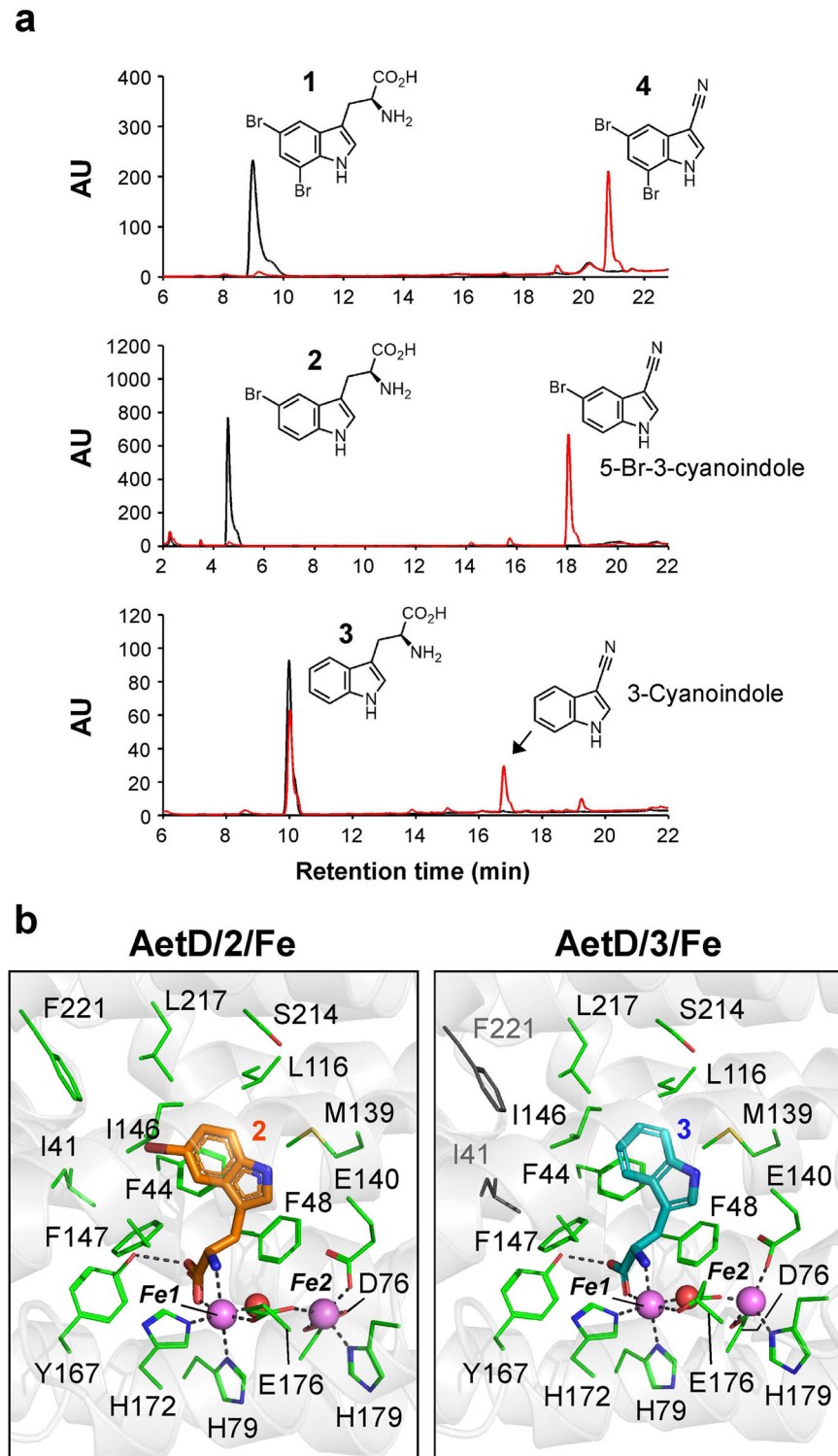

**Fig. 3 | AetD catalytic activity and complex structures of L-Trp with various degrees of bromination. a** The representative HPLC chromatograms of reactions containing three substrates in the presence (red traces) or absence of AetD (black traces). **b** The enzyme-substrate interaction networks of AetD complexes. The bound substrates, interacting residues, Fe ions, and water molecules are displayed as described in Fig. 2c. All residues that constitute the binding pocket of **1** as shown in Fig. 2c are displayed, with those distant from the bound ligands colored in gray (>5 Å, F221 and I41 in AetD/**3**/Fe). The dashed lines indicate distance <3.5 Å. The $2F_o$-$F_c$ and $F_o$-$F_c$ maps of **1**, Fe ions, and the coordinating waters and residues are shown in Supplementary Fig. 7.

catalyzed reaction is compared with other HO-like diiron enzymes. The currently characterized HO-like diiron enzymes can be classified into two groups based on their catalytic reactions: (1) lyases/desaturases, such as UndA and BesC[8,15] and (2) N-oxygenases including SznF (also named StzF)[16,17], RohS[18] and BelK/HrmI[19,20]. For the fatty acid

decarboxylase UndA, structural evidence indicates that the carboxyl O of lauric acid binds to Fe1, which was supposed to trigger $O_2$ addition to yield the peroxo-$Fe_2$(III/III) intermediate to facilitate subsequent decarboxylation (Supplementary Fig. 10a)[9]. The existence of the analogous peroxo-$Fe_2$(III/III) species in another desaturase BesC has also

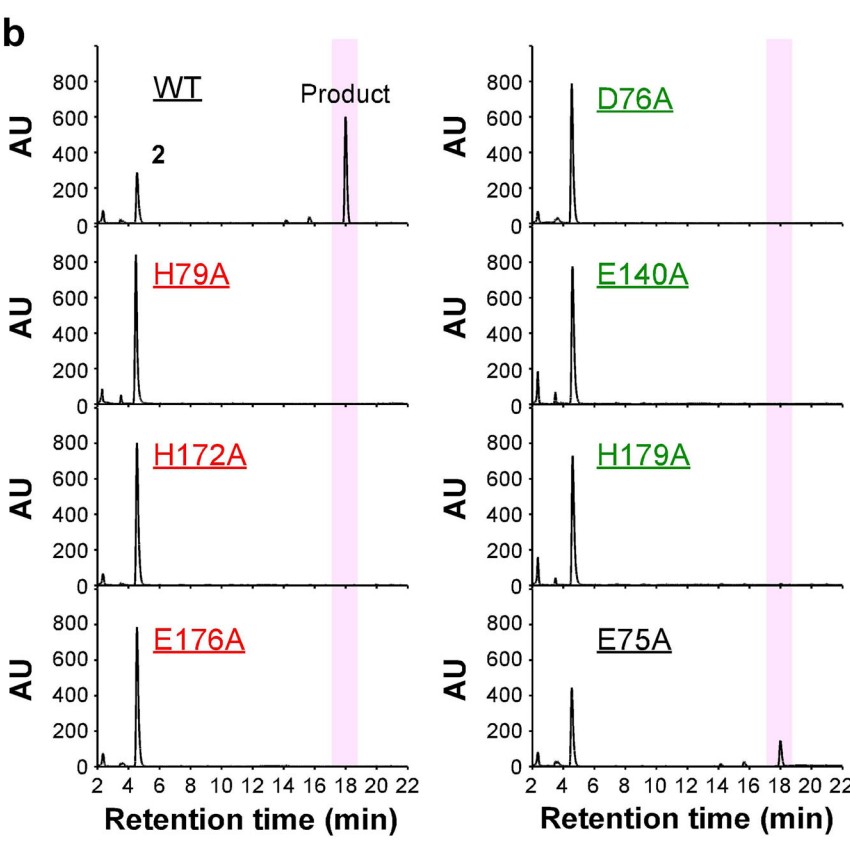

**Fig. 4 | Fe coordination sites of AetD. a** The Fe-binding sites in AetD/**1**/Fe and several HO-like structures including UndA from *P. fluorescenes* (PDB ID, 6P5Q), CADD from *Chlaymydia trachomatis* (PDB ID, 1RCW) and HO-like domain of SznF from *Streptomyces achromogenes* (PDB ID, 6VZY). The Fe-coordinating residues and ligands are shown in lines and sticks, respectively. Fe ions and water molecules are shown in pink and red spheres, with two Fe sites labeled numerically. Dashed lines, distance <3.5 Å. **b** The representative HPLC chromatograms of reactions containing **2** incubated with wild type or variant AetD that harbor Ala substitutions of potential Fe-binding residues. The time interval between 17-19 min when the reaction product was eluted is highlighted by pink columns.

been validated by spectroscopic analyses, which was proposed to trigger the radical formation on N7 or C4 of 4-Cl-L-lysine to afford 4-Cl-L-allylglycine[10] (Supplementary Fig. 10a). SznF employs the N-oxygenating peroxo-Fe$_2$(III/III) to catalyze consecutive oxidations of N$^\omega$-methyl-L-arginine (L-NMA)[17,21] (Supplementary Fig. 10b). Although only complex structure containing diiron cluster is available, the substrate-binding poses of the substrate of SznF with N1 then N2 orienting toward the peroxo species can only be proposed via molecular dynamics and QM/MM calculations[22]. Notably, residue E189 in SznF is located to the position of the substrate carboxyl in UndA coordinates Fe1 (Fig. 4a), a feature that has been proposed to relate to the lack of accelerated oxygen-addition upon the binding of substrate[22]. RohS and BelK/HrmI catalyze N-oxygenation to generate nitro group on 2-aminoimidazole and L-lysine, respectively (Supplementary Fig. 10a).

Compared with the known HO-like enzymes, the reaction catalyzed by AetD is apparently more complicated. Judging from the substrate-binding mode, both the carboxyl O and amino N of the substrate of AetD that are liganded to the diiron cluster could be oxidized (Supplementary Fig. 10c). Given that the carboxyl moiety of the substrate is depleted by the enzyme action, AetD should be able to catalyze decarboxylation, resembling UndA. As to the nitrile bond construction that might involve an unusual rearrangement process as previously proposed[2], it is suspected that the oxidation on the amino N of the substrate might play a role. Although it is difficult to resolve the comprehensive catalytic process of AetD at this stage, we proposed that multiple rounds of oxidation on the substrate should take place during the enzyme reaction.

Taken together, we report the complex structures of AetD, a HO-like enzyme that catalyzes an unprecedented nitrile formation reaction. Because AetD can transform L-Trp with various degree of bromination, it is expected to be capable of producing additional nitrile compounds that may serve as pharmaceutical or metabolic intermediates[23,24]. Altogether, these structural analyses and biochemical experiments are of fundamental importance to guide further investigations and applications of this unique nitrile synthase.

## Methods

### Chemicals
The crystallization screening kits were purchased from Hampton Research (Aliso Viejo, USA). The crystallization reagents were purchased from Sigma-Aldrich (St. Louis, USA). The chemical compounds utilized in enzyme-catalyzed reactions were purchased from J&K Scientific Ltd. (Beijing, CHINA), Acmec Biochemical Co., Ltd. (Shanghai, CHINA), and Bide Pharmatech Ltd. (Shanghai, CHINA).

### Preparation of 5,7-di-Br-L-Trp
5,7-Di-Br-L-Trp is prepared following the previously described protocol with some modifications[2]. Reaction mixture containing 3.2 µM recombinant AetF, 2 mM 5-Br-L-Trp (S93280, ACMEC, Shanghai, CHINA), 20 mM NaBr, 4 mM FAD and 4 mM NADPH in a buffer of 300 mM NaCl, 25 mM Tris-HCl [pH 7.5] in 20 mL was incubated at 30 °C for 8 h. The reaction was terminated by adding an equal volume of methanol, and centrifuged at 12,000 rpm for 10 min before extracted with EtOAc (3 × 20 mL). The organic layers were combined and washed with brine, dried with anhydrous Na$_2$SO$_4$ and concentrated in vacuo. The mixture was then separated by InerSustain C18 column (4.6 × 250 mm, 5 µm) with high-performance liquid chromatography system (HPLC, Shimadzu LC-20AD, JAPAN) equipped with a SPD-M20A photodiode array detector. Elution was performed with solvent A (double distilled water) and solvent B (methanol) with a linear gradient of 10−90% solvent B in 20 min. The fractions containing 5,7-di-Br-L-Trp were collected and concentrated in vacuo again to obtain a pure product.

### Gene cloning and mutagenesis
The genes encoding AetD (GenBank no. WP_208344496) and AetF (GenBank no. WP_208344498.1) fused with a linker containing a His$_6$ tag, Tobacco Etch Virus (TEV) protease cleavage site and (AG)$_5$ linker on the N-terminus were individually chemically synthesized and inserted into vector pET-32a for recombinant protein expression in *Escherichia coli* BL21(DE3). The amino acid sequences of the gene products are shown in Supplementary Table 1. The AetD variants were generated by polymerase chain reaction-based site-directed mutagenesis using pET-32a/AetD plasmid as the template. The mutagenesis oligonucleotides used for the construction of variants were listed in Supplementary Table 2. All plasmids were verified by direct sequencing.

### Recombinant protein expression and purification
The recombinant protein of AetD (wild type and variants) and AetF were expressed and purified following the same procedures. The recombinant plasmids were transformed into *E. coli* BL21(DE3) cells, grown in LB medium at 37 °C to an OD$_{600}$ of ~ 0.8 and induced by 0.3 mM isopropyl β-D-thiogalactopyranoside at 16 °C for 20 h. Cells were collected by centrifugation at 6000 × $g$ for 10 min and then resuspended in a lysis buffer containing 25 mM Tris-HCl [pH 7.5], 150 mM NaCl, and 20 mM imidazole, followed by disruption with a French press. Cell debris was removed by centrifugation at 17,000 × $g$ for 1 h. The supernatant was then applied to a Ni-NTA column with a fast protein liquid chromatography (FPLC) system (GE Healthcare) and eluted using an imidazole gradient from 20 to 300 mM and the target protein-containing fractions were collected. The protein solution was dialyzed against a buffer containing 25 mM Tris-HCl [pH 7.5] and 150 mM NaCl (300 mM NaCl for AetF), and subjected to the TEV protease digestion (final concentration, 5 µg mL$^{-1}$) overnight to remove the His$_6$ tag. The mixture was then passed through a Ni-NTA column again to remove the His-tagged portion, leaving the untagged protein eluted in the imidazole-free buffer. The protein purity (>95%) was verified by SDS-PAGE analysis. The proteins were quantified by OD280 measurement by using the extinction coefficients of 1.235 and 1.231 for AetD and AetF, respectively.

### Size exclusion chromatograph analysis
The molecular mass of the recombinant AetD in solution was estimated by size exclusion chromatographic analysis by using a Superdex 200 10/300 GL column with FPLC system (GE Healthcare). 100 µL solution containing 10 mg mL$^{-1}$ AetD in a buffer of 25 mM Tris-HCl [pH 7.5] and 150 mM NaCl was applied to the column and eluted with a flow rate of 0.2 mL min$^{-1}$. AetD was eluted at 14.3 mL which equals ~55.8 kDa based on the calculation from the protein markers.

### Crystallization and structure determination
All protein crystallization was conducted at 20 °C under aerobic conditions using the sitting-drop vapor-diffusion method. Prior to crystallization trial, 8 mg mL$^{-1}$ AetD-containing solution in a buffer containing 25 mM Tris-HCl [pH 7.5], and 150 mM NaCl was incubated with 5 mM indicated compounds for 30 min. Then 1 µL protein solution was mixed with 1 µL reservoir solution in 96-well Cryschem plates. The crystallization condition was optimized to 10 mM nickel chloride, 20% PEG 2000 MME, and 100 mM Tris [pH 8.5]. For no Fe crystals, crystals were soaked a cryoprotectant (mother liquid containing 15% ethylene glycol) prior to data collection. For Fe-containing crystals, crystals were soaked with a cryoprotectant containing 5 mM (NH$_4$)$_2$Fe(SO$_4$)$_2$ for 0.5 to 2 h prior to data collection. Notably, higher concentration of (NH$_4$)$_2$Fe(SO$_4$)$_2$ and longer soaking time could devastate the quality of diffraction such that careful examinations of multiple crystals were carried out to obtain satisfactory results. X-ray diffraction data were collected at beam line TPS05A of the National Synchrotron Radiation

Research Center (NSRRC, Hsinchu, Taiwan) and processed by using HKL2000[25]. The complex structure of AetD and 5,7-di-Br-L-Trp (**1**) was solved by the method of molecular replacement (MR) with a model predicted by AlphaFold2[26,27] as a search template. Subsequent model adjustment and refinement were conducted by using Refmac5[28], PHENIX[29], and Coot[30]. Prior to structure refinement, 5% of randomly selected reflections were set aside for calculating $R_{\text{free}}$ as a monitor of model quality. The other complex structures were solved by MR with AetD/**1**/Fe as a search model. All protein structure figures were prepared using the PyMOL program (http://pymol.sourceforge.net/). Data collection and refinement statistics are summarized in Table 1.

## Atomic absorption spectrometry
1 mg of the purified recombinant protein of AetD was added to 1 mL 69% nitric acid, heated until completely dissolved and diluted to 10 mL by ddH$_2$O. The samples were analyzed by using High-Resolution Continuum Source atomic Absorption spectrometer ContrAA700 (Analytik Jena, Jena, Germany) with a 234 W xenon short-arc lamp as a continuum radiation source. Using acetylene as the fuel and air as the oxidant and Aspect CS 2.0.0 software (Analytik Jena, Jena, Germany) was used to detect Fe ($\lambda = 248.3272$ nm) and Ni ($\lambda = 232.0030$ nm), respectively.

## Enzyme activity measurement
200 μL reaction mixtures containing 17.9 μM AetD, 1 mM Fe(NH$_4$)$_2$(SO$_4$)$_2$, 1 mM ascorbic acid, and 1 mM indicated substrates in 25 mM Tris-HCl buffer [pH 8.0] were incubated at 30 °C for 2 h. The reactions were terminated by adding an equal volume of methanol, passed through a 0.22 μm filter, and centrifuged at 12,000 rpm for 10 min before applied to a high-performance liquid chromatography system (HPLC, Shimadzu LC-20AD, JAPAN) equipped with a SPD-M20A photodiode array detector. Separation was conducted by using the InerSustain C18 column (4.6 × 250 mm, 5 μm) with a flow rate of the mobile phase 1.0 mL min$^{-1}$. When using 5-Br-L-Trp or 5,7-di-Br-L-Trp as a substrate, solvent A (double distilled water) and solvent B (acetonitrile) were used for elution with a linear gradient of 0−3 min 25% B, 3−23 min 25−100% B, 23−25 min 100% B and the eluent was monitored at 224 nm. When using L-Trp as a substrate, solvent A (double distilled water) and solvent B (methanol) were used for elution with a linear gradient of 10−90% B in 20 min and the eluent was monitored at 278 nm. The data of mass spectra (MS) was collected in a positive ion mode by an Agilent 6224 TOF LC-MS system (Agilent Technologies, Palo Alto, CA, USA).

## Data availability
The atomic coordinates and structure factors of the reported structures have been deposited in the Protein Data Bank under accession code as follows: AetD/**1**/Fe, 8JI3; AetD/**2**/Fe, 8JI5; AetD/**3**/Fe, 8JI7; AetD/**1**, 8JI2; AetD/**2**, 8JI4; AetD/**3**, 8JI6. The source data underlying Figs. 3a and 4b are provided as a Source Data file. Source data are provided with this paper.

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

## Acknowledgements

This work was supported by the National Key Research and Development Program of China (2021YFA0911400), Hubei Hongshan Laboratory (2022hszd030), the National Natural Science Foundation of China (32100711 and 82271887); the Natural Science Foundation of Hubei Province (2022CFA101 and 2022CFB360). We thank NSRRC (National Synchrotron Radiation Research Center, Taiwan) for access to beam line TPS-05A which contributed for the synchrotron data collection.

## Author contributions

H.L., H.Z., S.D., and Q.Z. carried out cloning, mutagenesis, protein purification, and crystallization. H.L. and Yunyun Yang measured the enzyme activity. H.L., J.-W.H., L.D., and Yu Yang collected crystallographic data, solved, refined, and analyzed the crystal structure. L.Y., J.M., R.-T.G., and C.-C.C. designed experiments, supervised the project, and prepared the manuscript.

## Competing interests

The authors declare no competing interests.
