## [Peer Review File · Nature Communications]

REVIEWER COMMENTS

Reviewer #1 (Remarks to the Author):

In this article, x-ray crystal structures are reported of the newly discovered nitrile synthase, AetD. This enzyme requires iron for activity and belongs to a newly emerging superfamily of iron-dependent oxygenases termed heme-oxygenase like diiron enzymes or HDOs. The work detailed in this manuscript includes structures of the enzyme with substrates and analogs bound. It is potentially significant because it could provide insight into the mechanism of the unusual reaction catalyzed by AetD, which is unique among HDOs.

However, the work has a major problem. The cofactor observed in the structures is modeled as a diiron center. But, in the materials and methods, there is no mention of specifically loading the protein with iron. Instead, the enzyme is exposed to high concentrations of Ni(II) at various stages in the purification and crystallization process. Most notably, the crystallization condition contains 10 mM Ni(II)chloride. Iron was added at the cryoprotectant stage. But the iron cofactors of HDOs are known to be extremely labile and challenging to incorporate. It seems extremely likely that the cofactor binding site is contaminated with Ni(II) – and there is no independent verification of the cofactor provided to validate the assignment of iron in the PDB files and mentioned throughout the manuscript. Until this issue is addressed either by doing additional experiments to validate the metal ion assignment (through collection of Fe and Ni anomalous diffraction datasets, spectroscopic characterization of the protein samples prior to crystallization, etc) – this manuscript cannot be published in its current form. And any resulting mechanistic information is called into question because the presence of the wrong metal could result in structural changes that might not be present in the catalytically relevant iron form of the enzyme.

Additional comments are provided below. But my opinion is that the above issue must be addressed first before any subsequent claims can be evaluated accurately.

Detailed comments:

Methods section:

1. In the gene cloning and mutagenesis section, tobacco is misspelled as “tabaco”
2. In the recombinant protein expression and purification section, Missing some details in cluding the amount of TEV protease used and the extinction coefficient used to calculate protein concentration.
3. In the size exclusion chromatography analysis section, use of the word “reaction” in the second sentence is unclear. Please revise.

Results section:

4. In the Crystal structure of AetD section, Third paragraph, sentence 2, UndA is misspelled as “UdnA” which happens again in the caption of Fig. S6
5. CADD no longer has unknown function or reactivity. Please read and cite the recent work by Allen and Makris on this topic.

Figures/tables

6. In general it would be helpful if the structure figures were larger.
7. The SznF sequence is missing three residues after E215 in this alignment. This makes it so that E215 and H225 of SznF appear closer together in sequence than they really are. It also changes how the SznF sequence aligns with other sequences. Please check all sequences and redo this alignment. Also, the way different residues were labeled was a little confusing to me, and labeling could probably be done in a more straightforward way.
8. S7: please explain which schemes are proposed and which you have evidence to support. There is crystallographic evidence to support the substrate coordinating the Fe cofactor in UndA, but not in SznF, like they show. The extra ligand in SznF may obviate substrate coordination to the cofactor, so they should probably alter this scheme to reflect that.
9. Table S1: the protein amino acid sequences start with HHHHHH. The starting Methionine and a short region would likely precede the hexahistidine tag. If so, please show this in the sequences.

Reviewer #2 (Remarks to the Author):

The submission by the Guo and Chen team on "The structural and functional investigation into an unusual nitrile synthase" is another valuable contribution to clarify the biosynthesis of AETX and its rather uncommon features, the bis-indole linkage, the high degree of bromination and the unusual nitrile group in one of the indole subunits. The work by this group provides clear evidence on the biocatalytic activity of the nitrile synthase (AetD). They solved the structure by co-crystallizing the protein with a suitable substrate.

All the results and discussions give a clear indication and insight into the robustness and reproducibility of the provided experimental details.

"applications of AetD as this enzyme might possess potentials in producing nitrile compounds from alternative amino acids" The authors state that the nitrile synthase catalyzes independent of the state of bromination, this led to the assumption to be suitable for other amino acids. The authors do not provide any proof for this, therefore, I would like this statement to be deleted (in introduction and conclusions). It will be very interesting to hear on these developments in the future, I am convinced the authors are already working on it. I very much enjoyed reading this submission, it is a very sound contribution. It meets the requirements of Nat.Comm., it is very attractive to the readership of this journal. I strongly recommend to publish this paper as is.

REVIEWER COMMENTS

Reviewer #1 (Remarks to the Author):

In this article, x-ray crystal structures are reported of the newly discovered nitrile synthase, AetD. This enzyme requires iron for activity and belongs to a newly emerging superfamily of iron-dependent oxygenases termed heme-oxygenase like diiron enzymes or HDOs. The work detailed in this manuscript includes structures of the enzyme with substrates and analogs bound. It is potentially significant because it could provide insight into the mechanism of the unusual reaction catalyzed by AetD, which is unique among HDOs.

However, the work has a major problem. The cofactor observed in the structures is modeled as a diiron center. But, in the materials and methods, there is no mention of specifically loading the protein with iron. Instead, the enzyme is exposed to high concentrations of Ni(II) at various stages in the purification and crystallization process. Most notably, the crystallization condition contains 10 mM Ni(II)chloride. Iron was added at the cryoprotectant stage. But the iron cofactors of HDOs are known to be extremely labile and challenging to incorporate. It seems extremely likely that the cofactor binding site is contaminated with Ni(II) – and there is no independent verification of the cofactor provided to validate the assignment of iron in the PDB files and mentioned throughout the manuscript. Until this issue is addressed either by doing additional experiments to validate the metal ion assignment (through collection of Fe and Ni anomalous diffraction datasets, spectroscopic characterization of the protein samples prior to crystallization, etc) – this manuscript cannot be published in its current form. And any resulting mechanistic information is called into question because the presence of the wrong metal could result in structural changes that might not be present in the catalytically relevant iron form of the enzyme.

Response: We thank the Reviewer for raising the question concerning the identity of the modeled metal ions in our structures. First of all, we should like to note that we literally “soaked” the AetD crystals in Fe ion-containing cryoprotectants for 0.5-2 hours instead of just dipped them in the cryoprotectant as the routine procedure for cryoprotection. In addition, it is also tricky to obtain good diffraction data, as soaking with a higher concentration of Fe or for a prolong time could lead to crystal cracking and devastate the diffraction quality. This could be attributed to the vulnerability of the diiron cofactor in HO-like proteins as the Reviewer mentioned. To clarify this point, we modified the section “Crystallization and structure determination” in **Methods**, which now reads “**For no Fe crystals, crystals were dipped in a cryoprotectant (mother liquid**

containing 15% ethylene glycol) prior to data collection. For Fe-containing crystals, crystals were soaked with a cryoprotectant containing 5 mM $(\text{NH}_4)_2\text{Fe}(\text{SO}_4)_2$ for 0.5 to 2 h prior to data collection. Notably, higher concentration of $(\text{NH}_4)_2\text{Fe}(\text{SO}_4)_2$ and longer soaking time could devastate the quality of diffraction such that careful examinations of multiple crystals were carried out to obtain satisfactory results.” It would be very difficult to distinguish Ni from Fe in the X-ray diffraction datasets because these two ions share highly similar molecular mass (Ni, 58.693 Da; Fe, 55.845 Da) and coordinating ligands (His and carboxylate residues). Performing anomalous diffraction might also yield confusing results because both Ni (in the crystallization buffer) and Fe ion (in the protein, please see below) were present in the crystals. So we decided to deal with this issue with alternative measures.

First, we agree with the Reviewer that the presence of Ni ion in the purification procedures may bind to the protein and complicates experimental results and spectroscopic characterization of the protein samples prior to crystallization should be performed. We conducted atomic absorption spectrometric analysis to detect the presence of Fe and Ni in the purified protein. As shown in **Fig. 1** (please see below), we detected Fe ion and no Ni ion in the purified AetD. Therefore, our protein purification procedures yielded recombinant AetD that only contained Fe ion and no Ni ion, which serves a reliable material for crystallization trial and biochemical experiments.

Fig. 1. Atomic absorption spectrometric analyses. The purified protein of AetD was subjected to atomic absorption spectroscopic analyses by monitoring 248.3272 nm and 232.003 nm wavelength to detect Fe and Ni ion, respectively.

Then, we tried to grow AetD crystals in crystallization solution free of Ni ion. Unfortunately, no crystals that yield X-ray diffraction datasets were available in spite of intensive efforts. Nonetheless, we omitted the Fe soaking step and successfully solved AetD crystal structures in complex with three substrates. These -Fe crystals

harbor a metal ion that is coordinated as the Fe1 site observed in the +Fe crystals (**Fig. 2**, see below). Based on the fact that the purified AetD protein carries Fe ion and the highly identical coordination status to other HO-like proteins, this metal ion was modeled with a Fe ion. Unlike the Fe1 site, the Fe2 site in -Fe crystals was vacant and the Fe2-coordination networks constituted by D76, E140 and H179 were broken (**Fig. 2**). Structural investigations clearly indicate that this is mainly owing to the disordered helix $\alpha 6$ that deviated H179 away from the catalytic center (**Fig. 3**). Because +Fe crystals and -Fe crystals share the same unit cell and space group, the conformational changes in helix $\alpha 6$ is not resulted from the altered crystal packing. Instead, it is possible that the Fe ion in the soaking trial binds to the enzyme and stabilizes H179 to constitute the intact diiron cluster. Notably, a metal ion that is located distant from the Fe2 site and coordinated by waters, a TRIS and/or residues including E140 and H72 was observed in -Fe crystals (**Fig. 2**). This metal was modeled with a Ni ion, whose presence could be competed by soaking crystals with Fe ion. From these results, we suspected that helix $\alpha 6$ stabilization, at least partially, might promote the growth of the AetD crystals, and this could be achieved through the formation of the diiron cluster or Ni ion.

Fig. 2. Metal ion coordination status in AetD complex structures. The enzyme-ligand interaction and metal ion coordination networks in AetD complex structures are displayed. Chain A in +Fe crystals and both chains (chain A and chain B) in -Fe crystals are shown. The bulge regions in helix $\alpha 6$ in chain A of -Fe crystals are traced by dashed curves, whose counterparts in chain B were not observed owing to the lack of electron density. The $2F_o - F_c$ electron density maps of protein residues, bound ligands, waters and metal ions are contoured at 2.0σ (red mesh) and 1.0σ (lightteal mesh).

Fig. 3. The Fe2 site dissociated in AetD crystals without loaded with Fe ion. The overall structure of chain A of AetD in complex with 5,7-di-Br-L-Trp (sticks noted by 1) with or without soaking with Fe ion. Note that Fe2 coordinating residue H179 is located on a bulge within the helix $\alpha 6$ region and deviated from the catalytic center.

Altogether, we concluded that (1) the purified AetD carries Fe ion and no Ni ion; (2) soaking AetD crystals with Fe ion resulted in the formation of the intact diiron cluster, which is assembled in a fashion the same as those in other HO-like proteins. We greatly appreciate the Reviewer for raising the question that prompts us to further validate the identity of the observed diiron cluster of AetD. We are convinced that these additional experimental results should provide concrete evidences to support our conclusions. We renamed the complexes containing a diiron cluster to AetD/1/Fe, AetD/2/Fe and AetD/3/Fe, and named those without soaked with Fe as AetD/1, AetD/2 and AetD/3 for clarity (please see revised **Table 1** for structural statistics of the newly added datasets). The descriptions regarding atomic absorption spectrometric analyses and crystals soaked with or without Fe ion have been added to the 2nd paragraph of AetD and substrate interaction network section of **Results**, which now reads “**The type of metal ions modeled in the AetD/1/Fe structure could be complicated by the presence of Ni ion in the purification procedures (nickel column) and crystallization buffer. We thus performed atomic absorption spectrometry and detected signal of Fe ion and not for Ni ion in AetD protein (Fig. S4), indicating that the purified AetD protein only carries Fe ion. Then we tried to grow crystals in buffers free of Ni ion but the efforts eventually failed. Nonetheless, we successfully solved the structure of AetD/1 crystal that was not soaked with Fe ion. The AetD/1 structure harbors the substrate and a metal ion that is coordinated and located as the Fe1 observed in AetD/1/Fe (Fig. S5). Based on the fact that the purified AetD carries Fe ion and the highly identical coordination**

status to other HO-like proteins, this metal ion should be reliably modeled with a Fe ion. Intriguingly, the Fe²⁺-corresponding site is vacant and the Fe²⁺-coordination networks are dissociated owing to the disordered helix α_6 that deviates H179 away from the Fe²⁺ site (**Fig. S5**). Because AetD/1 and AetD/1/Fe share the same unit cell and space group, the conformational change of helix α_6 is not a result of alternative crystal packing. Instead, it is possible that Fe ion in the soaking trial could bind to the enzyme and stabilize H179 to constitute the intact diiron cluster. Notably, a metal ion that is located distant from the Fe²⁺ site and coordinated by waters, a TRIS and/or residues including E140 and H72 was observed in AetD/1 (**Fig. S5**). This metal was modeled with a Ni ion, whose presence could be competed by soaking crystals with Fe ion.”. The descriptions regarding the complex structures of compound **2** and **3** without soaked with Fe have also been added to the end of the section AetD-catalyzed reaction is independent of bromination of **1** in **Results**, which now reads “In addition, the structures of these complex crystals without soaking with Fe ion were also solved (**Table 1**). These complexes harbor the same metal ion coordination statuses as that was observed in the AetD/1 structure (**Fig. S8**), which provide additional evidences to support the conclusions regarding the formation of the diiron cluster in AetD/Fe complex crystals.”

Additional comments are provided below. But my opinion is that the above issue must be addressed first before any subsequent claims can be evaluated accurately.

Response: Please see above.

Detailed comments:

Methods section:

1. In the gene cloning and mutagenesis section, tobacco is misspelled as “tabaco”

Response: Corrected.

2. In the recombinant protein expression and purification section, Missing some details including the amount of TEV protease used and the extinction coefficient used to calculate protein concentration.

Response: We thank the Reviewer for pointing out the missing information in Methods. 5 $\mu\text{g mL}^{-1}$ TEV protease was used for tag cleavage, and the extinction coefficients for AetD and AetF are 1.235 and 1.231, respectively. These information has been added to the referred section in the revised manuscript.

3. In the size exclusion chromatography analysis section, use of the word “reaction” in the second sentence is unclear. Please revise.

Response: We thank the Reviewer for pointing out this erroneous presentation and have corrected the referred phrase to “100 μL solution containing 10 mg mL^{-1} AetD”.

Results section:

4. In the Crystal structure of AetD section, Third paragraph, sentence 2, UndA is misspelled as “UdnA” which happens again in the caption of Fig. S6

Response: Corrected.

5. CADD no longer has unknown function or reactivity. Please read and cite the recent work by Allen and Makris on this topic.

Response: We thank the Reviewer for providing the update information regarding the function of CADD. The descriptions about the function of CADD and its unusual heterobimetallic Fe:Mn cofactor have been added to the 3rd paragraph in section Crystal structure of AetD of **Results**, which reads “UndA and BesC are oxidases that harbor a diiron cluster as a catalytic center to exercise decarboxylation and C-C bond cleavage, respectively^{9,10}, whereas **TenA is a protein with unknown function. CADD has been recently confirmed as a *p*-aminobenzoate synthase that hires a heterobimetallic Fe:Mn cofactor to exhibit its optimal activity^{12,13}, although a diiron cluster was observed in the enzyme in crystallographic analyses¹¹.”**

Figures/tables

6. In general it would be helpful if the structure figures were larger.

Response: We thank the Reviewer for the suggestion and have increase the size of the structural drawings.

7. The SznF sequence is missing three residues after E215 in this alignment. This makes it so that E215 and H225 of SznF appear closer together in sequence than they really are. It also changes how the SznF sequence aligns with other sequences. Please check all sequences and redo this alignment. Also, the way different residues were labeled was a little confusing to me, and labeling could probably be done in a more straightforward way.

Response: We thank the Reviewer for pointing out the problematic presentation of the sequence alignment. It is very difficult to align the Fe-coordinating residues of these HDOs in spite of their highly identical positions in the structures. We thus modified the figure as well as related descriptions in compliance with the Reviewer’s suggestions. In the revised manuscript, protein sequences of AetD, UndA, CADD and SznF are listed individually with their Fe coordinating residues highlighted (please see below for **Fig. S9**). The related description in the first paragraph in section Diiron cluster of AetD and

implications in catalytic mechanism of **Results** have also been modified accordingly, which now reads “In stark contrast to the low sequence identity (< 11%), the Fe-coordinating statuses in these HO-like proteins are highly conserved (**Fig. 4a**). Intriguingly, although Fe1 is coordinated by two His and one Glu, their order of appearance in the primary sequence of AetD (H-H-E) is different from those in other HO-like proteins (E-H-H) (**Fig. S9**). To further validate the structural observations, we thus conducted mutagenesis experiments...”

						The order of appearance of Fe coordinating residues		
AetD	10	20	30	40	50	60	Fe1	Fe2
MKAILQLILE	KRQEFKLP	FEFVRDETIS	PEERLILYPC	IAAFALNFRD	LNRYDYRDDN			
70	80	90	100	110	120			
SSDYYQKIIN	IHTQEDAKHW	EWFLNDLELL	GFDKTMRFSE	ALRFVWSDDL	LHTRRLCHNI		H-H-E	D-E-E-H
130	140	150	160	170	180			
AVLSHDLEPV	MKMVVIEAME	TAGLVIFHAL	AKPGESIACA	TRRKYLYVAD	SHVEVE TGHA			
								
UndA	10	20	30	40	50	60		
MIDTFSRTGP	LMEAAASPAP	TQQLIQDCSE	SKRRVVEHEL	YQRMRDNKL	AKVMRQYLIG			
70	80	90	100	110	120			
GWPVVEQFAL	YMAQNLTKTR	FARHPGEDMA	RRWLMRNIRV	ELNHADYVWH	WSRAHGVTLE		E-H-H	E-H
130	140	150	160	170	180			
DLQAQQVPPE	LHALSHWCWH	TSSADSLIVA	IAATNYAIEG	ATGEWSALVC	SNGIYAAAFP			
190	200	210	220	230	240			
EEDRKRAMKW	LKMHAQYDDA	HPWEALEIIV	TLAGLNPTKA	LQAE LRQAIC	KSYDYMYLFL			
								
CADD	10	20	30	40	50	60		
MMEVFMNFLD	QLDLIIQNKH	MLEHTFYVKW	SKGELTKEQL	QAYAKDYVLH	IKAFPKYLSA			
70	80	90	100	110	120			
IHSRCDDLEA	RKLLLDNLM	E ENGYPNHID	LWKQFVFALG	VTPEELEAHE	PSEAAKAKVA		E-H-H	E-E-D-H
130	140	150	160	170	180			
TFMRWCTGDS	LAAGVAALYS	Y ESQIPRIAR	EKIRGLTEYF	GFSNPEDYAY	FTEHHEADVR			
190	200	210	220	230				
HAREEKALIE	MLLKDDADKV	LEASQEVTS	LYGFLDSFLD	PGTCCSCHQS	Y			
								
SznF	190	200	210	220	230	240		
QFAPDFLSEA	SPMMRNVLGY	YGPAQSEWFK	VVID E YGYGV	HDTKHSTLFE	RTLESVGLS			
250	260	270	280	290	300			
DLHRYWQYYL	NSSLLLNMYF	HYLGKNHELF	FRYVGALYYT	E SSLVDFCRR	ADHLLREVFG		E*-E-H-H	E-E-D-H
310	320	330	340	350	360			
DTVDTTYTFE	H IHI D QHGR	MAREKIIKPL	VEAHGDGIIP	EIVRGIEEYR	VLLEIGDFDF			

Fig. S9 Partial protein sequences of AetD and several HO-like fold proteins. Partial protein sequences of AetD, UndA from *Pseudomonas* (PDB ID, 6P5Q), CADD from *C. trachomatis* (GenBank accession no. WP_009871978) and the HO-like domain of SznF from *S. achromogenes* (PDB ID, 6VZY) that contain Fe ion-coordinating residues are displayed. Residues serve to coordinate the Fe1 and Fe2 are colored in red and green, respectively. For E176 in AetD, E81 in CADD and E215 in SznF that participate in both Fe1 and Fe2 coordination, red-colored characters with green underlines are used for the labeling. E*, the SznF-unique E189, which is absent in other HDOs.

8. S7: please explain which schemes are proposed and which you have evidence to support. There is crystallographic evidence to support the substrate coordinating the Fe cofactor in UndA, but not in SznF, like they show. The extra ligand in SznF may obviate

substrate coordination to the cofactor, so they should probably alter this scheme to reflect that.

Response: We appreciate the Reviewer's suggestions and have modified the figure (please see below) and added more information regarding the source of the mechanism in the legend. Furthermore, the possible effects posed by the additional Fe1-coordinating E189 on the substrate binding of SznF have also been noted in the 2nd paragraph in section Diiron cluster of AetD and implications in catalytic mechanism of **Results** in the revised manuscript, which now reads "Although only complex structure containing diiron cluster is available, the substrate-binding poses of the substrate of SznF with N1 then N2 orienting toward the peroxo species has been proposed via molecular dynamics and QM/MM calculations²¹. Notably, the substrate-binding pattern of SznF could be more complicated than anticipated because the enzyme harbors an additional Fe1-coordinating residue (E189, **Fig. 4a**) that was proposed to obviate the binding of substrate¹⁰. Therefore, structural evidences to validate the substrate-binding pattern of SznF should be required."

Fig. S7 Catalytic reaction of HO-like diiron enzymes. The mechanism of UndA was proposed based on crystallographic analyses¹, while those of BesC and SznF were proposed based on spectroscopic analysis² and molecular dynamics and QM/MM calculations³, respectively. L-NMA, N^ω-methyl-L-arginine; L-HMA, N^δ-hydroxy-N^ω-methyl-L-Arg; L-DHMA, N^δ,N^ω-dihydroxy-N^ω-methyl-L-Arg.

9. Table S1: the protein amino acid sequences start with HHHHHH. The starting Methionine and a short region would likely precede the hexahistidine tag. If so, please show this in the sequences.

Response: We cloned the AetD- and AetF-encoding sequences to pET32a vector that carries a thioredoxin 1-encoding gene, His₆ tag, thrombin cleavage site, S-tag and enterokinase site. Therefore, the methionine that appears on most 5'-end belongs to thioredoxin 1. Behind these components, an additional His₆ tag followed by a TEV cleavage site were added in front of the target protein-encoding gene. Only the inserted gene fragments were shown in the previous version of manuscript, and the entire protein sequences from Met to the stop codon are now displayed in the revised **Table S1**.

Table S1. Amino acid sequences of recombinant AetD and AetF used in this study

AetD	MSDKIIHLTDDSFDTDVLKADGAILVDFWAEWCGPCKMIAPIL DEIADEYQGKLTVAKLNIDQNPGTAPKYGIRGIPTLLLKNGEV AATKVGALSKGQLKEFLDANLAGSGSGHMHHHHHSSGLVPR GSGMKETAATAAKFERQHMDSPDLGTDDDDDKAMEHHHHHHEN LYFQAGAGAGAGAGAGMKAILQLILEKRQEFEKLPCEFVRDETISP EERLILYPCIAAFALNFRDLNRYDYRDDNSSDYQKIINIHTQED AKHWEWFLNDLELLGFDKTMRFSEALRFVWSDDLHTRRLC HNIAVLSDLEPVMKMVVIEAMETAGLVIFHALAKPGESIAKAT RRKYLYVADSHVEVETGHAVGTENIITILEQTQLSSEQEEKAKEI VNKVFQWSTNLIGEFERYVKAHRSEKAQPTAAY
AetF	MSDKIIHLTDDSFDTDVLKADGAILVDFWAEWCGPCKMIAPIL DEIADEYQGKLTVAKLNIDQNPGTAPKYGIRGIPTLLLKNGEV AATKVGALSKGQLKEFLDANLAGSGSGHMHHHHHSSGLVPR GSGMKETAATAAKFERQHMDSPDLGTDDDDDKAMEHHHHHHEN LYFQAGAGAGAGAGAGMLEVCIIGFGFSAIPLVRELARTQTEFQIIS AESGSVWDRLSESGRLDFSLVSSFQTSFYSDLVRDYEKDYIPT AKQFYEMHERWRSVYEEKIIRDFVTKIENFKDYSLISTRSGKTY EAKHVVLATGFDRLMNTFLSNFDNHVSNKTFVFDTMGDSANL LIAKLIPNNKIILRTNGFTALDQEVQVLGKPFTLDQLESPNFRY VSSELYDRLMMSPVYPRTVNPAVSYNQFPLIRRFDSWVDSKSSP PNGLIAIKYWPIDQYYYHFNDLENYISKGYLLNDIAMWLHTG KVILVPSDTPINFDKKTITYAGIERSFHQYVKGDAEQPRLPTILIN GETPFEYLYRDTFMGVIPQRLNNIYFLGYTRPFTGGLANITEMQ SLFIHKLITQPQFHQKIHQNLSKRITAYNQHYGAAKPRKHDHT VPFGFYTEDIARLIGIHYQPNECRSVRDLLFYAFPNNAFKYRL KGEYAVDGVDELIQKVNDKHDHYAQVVFVQALSIRNMNSDEAA EWDHSARRFSFNDMRHKEGYRAFLDTYLKAYRQVENISVDDT VVDEEWNFMVKEACQVRDKVAPNIEEKTHYSKDEDVNKGIRL

	ILSILDSDISSLPDSNGSRGSGNLKEGDRLCKFEAQSIEFIRLLQ PKNYELLFIRESTVSPGSHRHGETA
--	--

Green, thioredoxin 1; orange, thrombin site; blue, S-Tag; purple, enterokinase site; red, His₆ tag; underlined, tabaco etch virus cleavage site; reduced-size, (AG)₅ linker; shaded, AetD and AetF.

Reviewer #2 (Remarks to the Author):

The submission by the Guo and Chen team on "The structural and functional investigation into an unusual nitrile synthase" is another valuable contribution to clarify the biosynthesis of AETX and its rather uncommon features, the bis-indole linkage, the high degree of bromination and the unusual nitrile group in one of the indole subunits. The work by this group provides clear evidence on the biocatalytic activity of the nitrile synthase (AetD). They solved the structure by co-crystallizing the protein with a suitable substrate.

All the results and discussions give a clear indication and insight into the robustness and reproducibility of the provided experimental details.

"applications of AetD as this enzyme might possess potentials in producing nitrile compounds from alternative amino acids" The authors state that the nitrile synthase catalyzes independent of the state of bromination, this led to the assumption to be suitable for other amino acids. The authors do not provide any proof for this, therefore, I would like this statement to be deleted (in introduction and conclusions). It will be very interesting to hear on these developments in the future, I am convinced the authors are already working on it.

Response: We appreciate the Reviewer's advices and have modified the referred descriptions accordingly. The last sentence in the **Introduction** has been rewritten, which now reads "These results should be of great importance to guide further mechanistic investigations of AetD.". The contents in the 10th line in Conclusions have been modified to "We also demonstrated that AetD can transform L-Trp with various degree of bromination, which suggests that this enzyme could produce additional nitrile compounds that may serve as pharmaceutical or metabolic intermediates^{22,23}.".

I very much enjoyed reading this submission, it is a very sound contribution. It meets the requirements of Nat. Commun., it is very attractive to the readership of this journal. I strongly recommend to publish this paper as is.

Response: We sincerely thank the Reviewer for the efforts in reviewing this manuscript and the positive comments on our works.

REVIEWERS' COMMENTS

Reviewer #1 (Remarks to the Author):

In this revised manuscript, the authors included additional discussion of the rationale for modeling metal ions in the active site of their x-ray structures of the enzyme AetD. As detailed below, I still have concerns that their crystals could contain heterogeneous mixtures and the presentation of results does not sufficiently acknowledge this possibility. Additionally, the discussion section of the manuscript lacks important citations and contains some inaccuracies that must be addressed prior to publication. Finally, while the work is purported to provide mechanistic insight into the AetD reaction, it remains unclear exactly what that new insight might be beyond simply defining the metalloenzyme class involved. If that's the key insight - that is fine - but I would recommend clearly stating that fact rather than including broad statements of significance or vague statements about how many "oxidations" are involved when there isn't any further detail provided about how the reaction might work.

Line 30, page 2. In the abstract, it would be ideal to detail exactly how the results provide insight into the reaction mechanism.

Line 93, page 5. The missing region (180-187) appears to be quite close to the active site. I think this proximity deserves more comment and analysis. Could it be related to heterogeneity in metal ion composition and oxidation state in the active site?

Lines 128-147. In this section describing the modeling of the metal ions, it needs to be explicitly acknowledged in the main manuscript that the crystallization condition contains large concentrations of Ni(II). And I remain unconvinced that the metal ions can be unambiguously assigned as iron without collection of anomalous diffraction data. I think a statement needs to be included that acknowledges this deficiency in the experimental work. And I also think a statement needs to be included that states that while certain binding sites are modeled in the PDB files as iron or nickel - they could contain heterogeneous mixtures and the model may be inaccurate for that reason. I also think it needs to be acknowledged that the oxidation state of any iron present could also be heterogeneous due to the mixture of pre-bound (probably oxidized) iron and added Fe(II). Additionally, it should be stated whether oxygen was present during the crystallization and soak procedures. These details are particularly important if the primary significance of this work is to understand the reaction mechanism. It is essential to know the oxidation states and identities of the metal to interpret details like coordination environment, substrate binding mode, etc.

Line 170-173. How exactly do the observations in the complexes of AetD/Trp solved without Fe soaking provide information about formation of the diiron cluster? As mentioned above, without conclusive information about the metal content of each site in the cluster obtained by crystallographic analysis of metal occupancy at each site - it isn't even proven that these structures even contain iron at both sites.

Line 179. Please cite Makris et al. JACS 2021 143 21416 in addition to ref. 10.

Line 180. Replace "statuses" with "ligands"

Lines 181-183. It would make more sense to discuss the "order of appearance of in primary sequence" of the ligands in terms of sequence alignment to the other well-characterized HDOs.

Lines 188-215. The connection between this paragraph and the structural insights from the work reported here on AetD are really not clear. This paragraph should be reorganized so that the connections are more clear. At this stage, I think the structures reported in this work provide confirmation that AetD belongs to the same family as other HO-like diiron enzymes. So I would recommend organizing the paragraph around that idea. Beyond that, I did not understand how the details provided about the other enzymes connect to what we learn in this work about the AetD

reaction.

Line 201-203. In SznF, the substrate modeling effort only provided insight into potential interactions between the L-NMA alpha-amine and carboxylate functional groups and a substrate binding motif (D185) near the active site. This result was experimentally validated by mutagenesis of D185. The modeling effort was not intended (nor did it actually) provide any insight into how the two different side chain N-atoms are positioned relative to the iron cofactor to explain how they are sequentially hydroxylated. Please revise or omit this section of the text and revise figure S10b to reflect that we do not actually know how the N-atoms of the side chain of the SznF substrate interact with the intermediate.

Lines 204-207. Why is ref. 10 cited here? The observation that SznF has an additional ligand came from ref. 21. Ref. 10 has nothing to do with that observation. Also it is not clear why the presence of an additional ligand would make substrate binding "more complicated." I would argue that substrate binding is actually more complicated in the desaturase-lyase enzymes because these systems require substrate for assembly of the diiron cluster (see ref. 10 for a discussion) or for addition of oxygen – whereas the N-oxygenases (SznF included) do not require substrate to form the reactive diiron(III)-peroxo complex and instead likely bind substrate after intermediate formation.

Line 207. While it is true that it would be ideal to characterize the Fe(II/II)-SznF-substrate complex experimentally – this is likely not possible. In Bollinger et al. JACS 2020 142 11818 – it was noted that SznF does not require substrate in the Fe(II/II) state to react with O₂ – and even after formation of the reactive intermediate – the binding constant for L-NMA is in the mM regime. Therefore it is likely impossible to solve a structure of the Fe(II/II) complex with substrate – and very difficult to assemble a complex with substrate in the Fe(III/III)-peroxo state because that intermediate is very fleeting in SznF. So it is not really accurate to say that it is "required" to determine a structure with substrate in this system. It is actually impossible because of the relative kinetics and order of addition of substrates. Please remove this statement.

Lines 211-215. It is not clear what actual new insight into reaction mechanism is provided by this work. Why should "multiple rounds of oxidation...be expected?" Oxidation of what? The iron? Or the substrate? And what is the "identity of the peroxo species?" While certain intermediates are referenced in this paragraph for other HO-like diiron systems – no links were made to AetD – so it is not clear what "peroxo species" is being referenced here.

Lines 224-225. I don't think it's fair to say that rearrangements have not been reported for any HO-like diiron enzyme. In Li et al. Science 2021 374 1005, a rearrangement reaction was reported for a predicted HO-like diiron enzyme.

Figures S3, S5, S7, and S8. Please use difference maps rather than 2Fo-Fc maps to show the electron density for the substrate, ligands, metals etc. This is standard practice and more unbiased than the current representation.

Figure S10. The reaction mechanism for BesC is missing the ammonia and formaldehyde coproducts. Also – the substrate likely binds via the alpha amine and carboxylate rather than the side chain epsilon amine. Please see ref. 10 for substrate analog experiments that support this hypothesis.

Reviewer #3 (Remarks to the Author):

The authors addressed very nicely the comments of the reviewers. The question about whether Fe or Ni is located within the crystals structure is indeed very tricky and difficult to sign with 100%

certainly.

By using atom absorption spectroscopy it is nicely shown that the protein is loaded with Fe prior to crystallisation. Secondly crystal structures were obtained without adding additional Fe which yielded structure where one of the Fe molecules is not present instead a Tris molecule and a Ni molecule is present. Furthermore the side chain of amino acid E140 moves away. This all indicated to my understanding nicely that the Fe is bound in the structure. Therefore I justify the publication of the manuscript.

One comment to the authors.

Since you have such nice X-ray data I wonder if a justification of the Iron can also be concluded by looking at the B factors of the Fe ion as well as the side chains around it. Here the B factors should be significantly lower in the Iron containing structure. This information needs to be included in the paper. This would even make the conclusion stronger and would be next to atom absorption microscopy which deals with the protein prior to crystallisation be a measure which can be seen within the crystal structure. Please include this in the manuscript.

Reviewer #1 (Remarks to the Author):

In this revised manuscript, the authors included additional discussion of the rationale for modeling metal ions in the active site of their x-ray structures of the enzyme AetD. As detailed below, I still have concerns that their crystals could contain heterogeneous mixtures and the presentation of results does not sufficiently acknowledge this possibility. Additionally, the discussion section of the manuscript lacks important citations and contains some inaccuracies that must be addressed prior to publication. Finally, while the work is purported to provide mechanistic insight into the AetD reaction, it remains unclear exactly what that new insight might be beyond simply defining the metalloenzyme class involved. If that's the key insight - that is fine - but I would recommend clearly stating that fact rather than including broad statements of significance or vague statements about how many "oxidations" are involved when there isn't any further detail provided about how the reaction might work.

Response: We sincerely thank the Reviewer for reviewing the submitted manuscript again and herein submit a revised version of manuscript. We agree that it is still very difficult to comprehensively resolve the catalytic processes of AetD at this stage but are convinced that the structural information revealed in the manuscript should be of valuable. **First**, the substrate-binding pattern, the primary determinant of the enzyme catalytic reaction, is unambiguously revealed. **Second**, the enzymatic reaction is independent of bromination of the substrate, which indicates that AetD might possess a wide substrate spectrum. **Third**, the diiron cluster composition has been identified, which facilitate the further investigations into the molecular mechanism of AetD. **Fourth**, the amine and carboxylate of the substrate bind to the diiron cluster and are subjected to oxidation. We would also like to mention that two other Reviewers have approved the identification of the diiron cluster based on experimental evidences. Again, we appreciate the Reviewer's great efforts in evaluating our manuscript and provided many important comments that greatly improved the quality of the manuscript. Please see point-by-point responses for the raised questions below.

Line 30, page 2. In the abstract, it would be ideal to detail exactly how the results provide insight into the reaction mechanism.

Response: We appreciate the Reviewer's suggestions on providing more concise descriptions about the indications of the current study, and have modified the referred statements which now reads "Altogether, the present study reveals the substrate-binding pattern and validates the diiron cluster-comprising active center of AetD, which should provide important basis to support the mechanistic investigations into this new class of nitrile synthase."

Line 93, page 5. The missing region (180-187) appears to be quite close to the active site. I think this proximity deserves more comment and analysis. Could it be related to heterogeneity in metal ion composition and oxidation state in the active site?

Response: We thank the Reviewer for raising the question concerning the relevance between the enzyme reactivity and residue 180-187. Despite the 180-187 fragment is

located on helix $\alpha 6$ and positioned just behind H179, a Fe²⁺-coordinating residue that is essential to the catalytic reaction, we do not possess experimental evidences to support that any residue within this region plays a role in metal ion coordination, substrate-binding and catalytic reaction of AetD. We are also incapable of making any statement on the redox status of the active site. Despite we cannot provide more discussions on this event, we will pay attention to the conformational change of this region in the future works following the Reviewer's advice.

Lines 128-147. In this section describing the modeling of the metal ions, it needs to be explicitly acknowledged in the main manuscript that the crystallization condition contains large concentrations of Ni(II). And I remain unconvinced that the metal ions can be unambiguously assigned as iron without collection of anomalous diffraction data. I think a statement needs to be included that acknowledges this deficiency in the experimental work. And I also think a statement needs to be included that states that while certain binding sites are modeled in the PDB files as iron or nickel – they could contain heterogeneous mixtures and the model may be inaccurate for that reason. I also think it needs to be acknowledged that the oxidation state of any iron present could also be heterogeneous due to the mixture of pre-bound (probably oxidized) iron and added Fe(II). Additionally, it should be stated whether oxygen was present during the crystallization and soak procedures. These details are particularly important if the primary significance of this work is to understand the reaction mechanism. It is essential to know the oxidation states and identities of the metal to interpret details like coordination environment, substrate binding mode, etc.

Response: Although the Reviewer's concerns on the identity of the diiron cluster is understandable, we would like to mention that two other Reviewers have approved the results presented to support the presence of the diiron cluster in the structures in the revised manuscript. To further clarify our point, additional descriptions regarding the identity and status of the metal ions are provided in compliance with the Reviewer's suggestions. First we stated the obstacle in applying the anomalous diffraction technology to identify the metal species in the 3rd line of the 2nd paragraph in AetD and substrate interaction network in the Results section, which reads “**This event can be resolved by using anomalous diffraction technology to directly identify the species of the metal ion observed in crystals. However, it is hard to determine whether the signal of Ni is derived from the ions in the crystals or in the crystallization buffer.**”. In line 14 in the same paragraph, we also mention that a heterogeneous mixture of Fe and Ni ion, as well as Fe²⁺ and Fe³⁺, could be observed in the active site by the following sentences “**However, it should be noted that the possibility that a heterogeneous mixture containing Fe and Ni ion was observed in these crystals cannot be completely excluded. In addition, the presence of both Fe²⁺ and Fe³⁺ is possible although the crystallization and soaking trials were conducted under aerobic condition.**”. We also note the aerobic condition in Crystallization and structure determination section in the Methods, which reads “All protein crystallization was conducted at 20 °C **under aerobic condition** using the sitting-drop vapor-diffusion method.”

Line 170-173. How exactly do the observations in the complexes of AetD/Trp solved without Fe soaking provide information about formation of the diiron cluster? As mentioned above, without conclusive information about the metal content of each site in the cluster obtained by crystallographic analysis of metal occupancy at each site – it isn't even proven that these structures even contain iron at both sites.

Response: We thank the Reviewer for the question. The reason to justify the diiron cluster in AetD/Trp has been described in the section “AetD and substrate interaction networks” of the Results (please also see above) and shall not be repeated here.

Line 179. Please cite Makris et al. JACS 2021 143 21416 in addition to ref. 10.

Response: Done.

Line 180. Replace “status” with “ligands”

Response: Corrected.

Lines 181-183. It would make more sense to discuss the “order of appearance of in primary sequence” of the ligands in terms of sequence alignment to the other well-characterized HDOs.

Response: The Fe1 coordination ligands in AetD and other known HO-like enzymes comprise two His and one Glu protein residues. Despite these residues occupy the same spatial locations, their presence on the primary sequence in AetD and other HO-like enzymes are different. As a result, the order of appearance of Fe1-coordinating residues in the primary sequence in AetD and other HO-like enzymes are His-His-Glu and Glu-His-His, respectively. This is the reason that showing the Fe-coordinating residues by using a direct sequence alignment in the original version of the manuscript caused confusion and was suggested to be modified by the Reviewer. In the revised manuscript, the Fe-coordinating residues in the amino sequences of each protein are individually displayed to enable a direct comparison. To make our point clearer, a sentence was added after the referred descriptions, which reads “This complicates the identification of Fe1-coordinating residues only based on the sequence alignment.”

Lines 188-215. The connection between this paragraph and the structural insights from the work reported here on AetD are really not clear. This paragraph should be reorganized so that the connections are more clear. At this stage, I think the structures reported in this work provide confirmation that AetD belongs to the same family as other HO-like diiron enzymes. So I would recommend organizing the paragraph around that idea. Beyond that, I did not understand how the details provided about the other enzymes connect to what we learn in this work about the AetD reaction.

Response: We appreciate the Reviewer's advice and add a sentence to state that the following descriptions were made to discuss the possible reaction process of AetD based on the knowledge on the known HO-like enzymes, which reads “Based on the presented structural analyses and mutagenesis experiments, AetD should employ a

diiron cluster as a catalytic center, a feature shared by HO-like diiron enzymes. Therefore, the AetD-catalyzed reaction is compared with other HO-like diiron enzymes.”.

Line 201-203. In SznF, the substrate modeling effort only provided insight into potential interactions between the L-NMA alpha-amine and carboxylate functional groups and a substrate binding motif (D185) near the active site. This result was experimentally validated by mutagenesis of D185. The modeling effort was not intended (nor did it actually) provide any insight into how the two different side chain N-atoms are positioned relative to the iron cofactor to explain how they are sequentially hydroxylated. Please revise or omit this section of the text and revise figure S10b to reflect that we do not actually know how the N-atoms of the side chain of the SznF substrate interact with the intermediate.

Response: We thank the Reviewer for the suggestion and modify the referred sentence, which now reads “Although only complex structure containing diiron cluster is available, the substrate-binding poses of the substrate of SznF with N1 then N2 orienting toward the peroxo species **can only be** proposed via molecular dynamics and QM/MM calculations²².”. In addition, we marked panel that depicts SznF reaction in figure S10b with an asterisk and noted in the legend that the substrate-binding poses and reaction process of SznF are proposed based on molecular dynamics and QM/MM calculations.

Lines 204-207. Why is ref. 10 cited here? The observation that SznF has an additional ligand came from ref. 21. Ref. 10 has nothing to do with that observation. Also it is not clear why the presence of an additional ligand would make substrate binding “more complicated.” I would argue that substrate binding is actually more complicated in the desaturase-lyase enzymes because these systems require substrate for assembly of the diiron cluster (see ref. 10 for a discussion) or for addition of oxygen – whereas the N-oxygenases (SznF included) do not require substrate to form the reactive diiron(III)-peroxo complex and instead likely bind substrate after intermediate formation.

Response: We thank the Reviewer for the corrections on the descriptions regarding the Fe ion coordination status of SznF. What we intended to state is that the substrate-binding mode of SznF is distinct from those of other HO-like enzymes owing to its additional Fe1-coordinating protein residue. We agree with the Reviewer that the original statement is confusing and have modified the referred descriptions along with a correct literature. The text now reads “**Notably, residue E189 in SznF is located to the position of the substrate carboxyl in UdnA coordinates Fe1 (Fig. 4a), a feature that has been proposed to relate to the lack of accelerated oxygen-addition upon the binding of substrate²².**”

Line 207. While it is true that it would be ideal to characterize the Fe(II/II)-SznF-substrate complex experimentally – this is likely not possible. In Bollinger et al. JACS 2020 142 11818 – it was noted that SznF does not require

substrate in the Fe(II/II) state to react with O₂ – and even after formation of the reactive intermediate – the binding constant for L-NMA is in the mM regime. Therefore it is likely impossible to solve a structure of the Fe(II/II) complex with substrate – and very difficult to assemble a complex with substrate in the Fe(III/III)-peroxo state because that intermediate is very fleeting in SznF. So it is not really accurate to say that it is “required” to determine a structure with substrate in this system. It is actually impossible because of the relative kinetics and order of addition of substrates. Please remove this statement.

Response: We thank the Reviewer for the suggestion and have removed the statement accordingly.

Lines 211-215. It is not clear what actual new insight into reaction mechanism is provided by this work. Why should “multiple rounds of oxidation...be expected?” Oxidation of what? The iron? Or the substrate? And what is the “identity of the peroxo species?” While certain intermediates are referenced in this paragraph for other HO-like diiron systems – no links were made to AetD – so it is not clear what “peroxo species” is being referenced here.

Response: We thank the Reviewer for the comments. The exact catalytic processes of AetD cannot be thoroughly resolved with these presented data. As the structural information shall be of importance for the mechanistic investigations in the future, we understand that we can only deduce some catalytic features of AetD based on the current knowledge. To make this clearer, we modified the related descriptions now read “Compared with the known HO-like enzymes, the reaction catalyzed by AetD is apparently more complicated. Judging from the substrate-binding mode, both the carboxyl O and amino N of the substrate of AetD that are liganded to the diiron cluster could be oxidized (Fig. S10c). Given that the carboxyl moiety of the substrate is depleted by the enzyme action, AetD should be able to catalyze decarboxylation, resembling UndA. As to the nitrile bond construction that might involve an unusual rearrangement process as previously proposed², it is suspected that the oxidation on the amino N of the substrate might play a role. Although it is difficult to resolve the comprehensive catalytic process of AetD at this stage, we proposed that multiple rounds of oxidation on the substrate should take place during the enzyme reaction.”. We consider these descriptions should be appropriate to reflect the indications of the present works.

Lines 224-225. I don't think it's fair to say that rearrangements have not been reported for any HO-like diiron enzyme. In Li et al. Science 2021 374 1005, a rearrangement reaction was reported for a predicted HO-like diiron enzyme.

Response: We thank the Reviewer for providing this information and have modified the referred paragraph as following. “Taken together, we report the complex structures of AetD, a HO-like enzyme that catalyzes an unprecedented nitrile formation reaction. Because AetD can transform L-Trp with various degree of bromination, it is expected to be capable of producing additional nitrile compounds that may serve as pharmaceutical or metabolic intermediates^{23, 24}. Altogether, these structural analyses

and biochemical experiments is of fundamental importance to guide further investigations and applications of this unique nitrile synthase.”

Figures S3, S5, S7, and S8. Please use difference maps rather than $2F_o-F_c$ maps to show the electron density for the substrate, ligands, metals etc. This is standard practice and more unbiased than the current representation.

Response: We thank the Reviewer for the suggestions and have provided the F_o-F_c omit maps along with the original $2F_o-F_c$ maps in all relevant figures.

Figure S10. The reaction mechanism for BesC is missing the ammonia and formaldehyde coproducts. Also – the substrate likely binds via the alpha amine and carboxylate rather than the side chain epsilon amine. Please see ref. 10 for substrate analog experiments that support this hypothesis.

Response: We thank the Reviewer for the comment concerning the BesC-catalyzed reaction. The referred figure has been corrected by showing ammonia and formaldehyde as the reaction products of BesC. As to the substrate-binding pose, albeit alpha amine and carboxylate are essential moieties for the substrates to bind to the enzyme, the diiron species-mediated oxygenation/deprotonation is expected to occur on the side chain fragment of the substrate. Although the substrate-binding pattern is important for enzyme reaction especially for BesC that appears to accept promiscuous substrates, we feel that the oxidation reaction catalyzed by the diiron cluster shall be the main theme to be discussed in the referred paragraph. Thus, we intended to keep the current statements without adding more descriptions.

Reviewer #3 (Remarks to the Author):

The authors addressed very nicely the comments of the reviewers. The question about whether Fe or Ni is located within the crystals structure is indeed very tricky and difficult to sign with 100% certainly.

By using atom absorption spectroscopy it is nicely shown that the protein is loaded with Fe prior to crystallisation. Secondly crystal structures were obtained without adding additional Fe which yielded structure where one of the Fe molecules is not present instead a Tris molecule and a Ni molecule is present. Furthermore the side chain of amino acid E140 moves away. This all indicated to my understanding nicely that the Fe is bound in the structure. Therefore I justify the publication of the manuscript.

Response: We thank the Reviewer for the efforts in reviewing our manuscript and the approval on our works.

One comment to the authors.

since you have such nice Xray data I wonder if a justification of the Iron can also be concluded by looking at the B factors of the Fe ion as well as the side chains around it. Here the B factors should be significantly lower in the Iron containing structure. This information needs to be included in the paper. This would even make the conclusion stronger and would be next to atom absorption microscopy which deals with the protein prior to crystallisation be a measure which can be seen within the crystal structure. Please include this in the manuscript.

Response: We thank the Reviewer for the suggestions. Please see below table for the B factors of the Fe ions and the side chains of the Fe-coordinating residues in all structures presented in the submitted manuscript. We agree with the Reviewer that B factors would be a very useful parameter to describe the mobility of these components, but realize that the estimation can only be conducted under strictly controlled conditions. Therefore, the B factors of the Fe1 ion and Fe1-coordinating residues are used to draw a background reference since the construction of Fe1 site is the same in all structures. The B factors of Fe1 ion and the Fe1-chelating side chains (except for the H172 in AetD/2) in chain A in crystals containing diiron cluster are higher than those containing one Fe ion, but the scenario is opposite for those in chain B (please see below table). Notably, the same trend was observed for the B factors of the substrates bound to each structure.

The B factors of the Fe2-coordinating residue D76 are higher in both chains in structures that only carry the Fe1 ion. This suggests that D76 exhibits higher mobility when not liganded to the Fe2 ion, which is within our expectation. Nonetheless, the B factors of E140 in chain A in the complexes only containing Fe1 ion are lower than those containing diiron cluster (except for E140 in AetD/2), and the scenario is opposite in chain B. This mimics the background reference, indicating that the mobility of E140 might be not reduced when chelates the Fe2 ion. Similar phenomena are observed for E176 that chelates both the Fe1 and Fe2, that the B factors in chain A

in the structures containing one iron are lower than those containing two irons while the results are opposites in chain B. For H179, the side chain mobility appears lower in chain A in structures containing Fe1 ion than those containing two Fe ions. This event in chain B in all structures containing one Fe ion cannot be evaluated due to the lack of the electron density maps of H179.

Altogether, inconsistent results were obtained from analyzing B factors of the Fe ions and the Fe-coordinating residues, which we find difficult to provide clear conclusions. It is not uncommon to see unexpected results from evaluating the thermal motion, as B factor could be influenced by many factors such as the quality of the crystal and the resolution of the dataset. Not to mention that some iron-chelating residues could be stabilized by nearby residues/solvents even in the absence of the metal ion. It is truly a pity that we did not obtain better readouts. Nonetheless, we sincerely appreciate the Reviewer's very insightful suggestions on conducting these analyses.

Table. B-factors of the Fe ions and the coordinating residues in AetD complex structures

		AetD/1/Fe	AetD/2/Fe	AetD/3/Fe	AetD/1	AetD/2	AetD/3
Bound ligand		5,7-Di-Br-L-Trp, Fe	5-Br-L-Trp, Fe	L-Trp, Fe	5,7-Di-Br-L-Trp	5-Br-L-Trp	L-Trp
Fe ion B-factor (\AA^2)							
Chain A	Fe1	32.16	21.68	12.91	24.00	20.51	9.03
	Fe2	43.84	35.21	27.75	-	-	-
Chain B	Fe1	25.99	14.88	6.62	25.79	22.88	14.76
	Fe2	32.80	23.96	16.72	-	-	-
Side chains B-factor (\AA^2)							
Chain A/Fe1	H79	33.64	22.18	15.94	28.15	24.90	11.67
	H172	33.87	21.79	14.55	27.31	23.17	12.61
	E176*	49.73	40.11	32.22	35.63	31.90	20.54
Chain A/Fe2	D76	35.68	28.06	18.82	40.91	32.66	25.06
	E140	44.79	32.82	23.04	39.49	33.75	20.70
	H179	76.87	71.07	64.33	44.11	52.38	32.55
Chain B/Fe1	H79	27.64	13.45	9.35	29.47	26.64	17.26
	H172	29.71	17.74	9.23	28.06	24.33	16.30
	E176*	36.66	26.2	22.84	51.47	43.18	39.20
Chain B/Fe2	D76	31.83	22.76	14.92	38.41	36.17	35.88
	E140	41.63	29.63	21.81	56.43	48.76	32.50
	H179 [#]	43.74	45.54	28.79	-	-	-
Substrate B-factor (\AA^2)							
Chain A		32.38	23.42	15.11	27.07	22.13	13.01
Chain B		29.50	20.88	9.78	29.36	23.86	18.56

* E176 chelates both Fe1 and Fe2 in complexes contain diiron cluster.

[#] H179 is not visible in chain B of all complex crystals without soaking Fe.